# Influence of the Electrical Conductivity of the Nutrient Solution in Different Phenological Stages on the Growth and Yield of Cherry Tomato

**Tao Lu [1], Hongjun Yu [1,*], Tanyu Wang [1], Taoyue Zhang [1], Chenhua Shi [2] and Weijie Jiang [1]**

[1]  Institute of Vegetables and Flowers, Chinese Academy of Agricultural Sciences, Beijing 100081, China; lutao@caas.cn (T.L.); tanyuwang1989@163.com (T.W.); msforwtzp@163.com (T.Z.); jiangweijie@caas.cn (W.J.)
[2]  Shanghai Wintong Ecological Engineering Co., Ltd., Shanghai 200333, China; leena@wintong.com
*   Correspondence: yuhongjun@caas.cn; Tel.:+86-010-82108797

**Abstract:** Soilless cultivation is an important alternative to traditional agriculture and facilitates harvest by allowing for the precise control of plant nutrients to maximize the vegetable production of uniform fruits. Nutrient solution concentration is a critical factor affecting nutrient supply in soilless cultivation. Although some nutrient solution concentrations throughout the growth cycle for tomatoes have been developed, there are limited studies on nutrient solution concentrations at different phenological stages. Hence, we studied the effects of nutrient solution concentrations in different growth stages on the physiology, yield and fruit quality of cherry tomatoes with a previously developed nutrient solution formulation. The whole growth cycle of the tomato was divided into three stages which were irrigated with a nutrient solution with different electrical conductivities (ECs). A total of five treatments were set: CK (EC was 3.0 ms·cm$^{-1}$ for the 1st–3rd stage), T1 (EC was 1.5 ms·cm$^{-1}$ for the 1st stage, 3.0 ms·cm$^{-1}$ for the 2nd–3rd stage), T2 (EC was 1.5 ms·cm$^{-1}$ for the 1st stage, 3.0 ms·cm$^{-1}$ for the 2nd stage, 4.5 ms·cm$^{-1}$ for the 3rd stage ), T3 (EC was 1.5 ms·cm$^{-1}$ for the 1st–2nd stage, 3.0 ms·cm$^{-1}$ for the 3rd stage), and T4 (EC was 1.5 ms·cm$^{-1}$ for the 1st stage, 4.5 ms·cm$^{-1}$ for the 2nd–3rd stage). The results showed that the tomato plants treated with T2 and T4 had the strongest growth (with the highest plant height and leaf formation) as well as the best leaf photosynthetic performance (the chlorophyll content and the net photosynthetic rate were significantly increased). Additionally, the use of T2 and T4 significantly improved cherry tomato fruit quality as reflected by the significant promotion of total soluble solids by 9.1% and 9.8%, respectively, as well as by the improvement of maturity by 12.9% and 13.7%, respectively. Additionally, the yields for treatments T2 and T4 were increased by 7.3% and 13.4%, respectively, which was mainly due to the increase in single fruit weight. More importantly, nutrient solution EC management improved fertilizer use efficiency: the partial fertilizer productivity of T1, T2, and T4 was increased by 2%, 7% and 14%, respectively, while that of T3 was reduced by 7%. A comprehensive comparison showed that the ranking of the effect on production was T4 > T2 > T1 > CK > T3. Our results suggest that the regulation of EC in different growth stages affects the growth and yield characteristics of cherry tomatoes. This study may provide some references for further research to adjust the concentration of nutrient solutions to improve the utilization rate of fertilizer and fruit quality.

**Keywords:** nutrient solution concentration; soilless culture; electrical conductivity; photosynthesis; fruit quality; yield

## 1. Introduction

Currently, soilless cultivation is gaining popularity as a new type of intensive and efficient technology [1], which eliminates the dependence on soil, space and climatic conditions of traditional soil cultivation by providing the most suitable environment for root growth and development [2,3]. Compared to the conventional growing system, the soilless system results in earlier production and leads to more uniform fruits, which are preferred by the market [4]. In China, organic substrate cultivation has been widely

practiced and currently accounts for over 75% of the total area of soilless cultivation, which can increase the efficient use of land resources, save water and fertilizer, facilitate nutrient uptake, increase yield and protect the environment [5,6]. As one of the most important resources for the production of growing media in horticulture, coco peat is increasing in popularity as it is extracted from the outer hull of coconut and can be composted after use [7]. The plants are grown in this substrate with a constant supply of nutrient solution allowing for optimal control of mineral nutrition. Cherry tomato (*Solanum lycopersicum* var. *cerasiforme*) is a wild relative of cultivated varieties and originated in South America and Mexico [8]. Its fruits have a high content of sugars and health-promoting compounds and are convenient to consume fresh. Fruit quality, which is mainly determined by its genetics but is also affected by growing environmental conditions, including water and nutrients, is one of the key factors affecting tomatoes' economic benefits [9]. Tomato fruit quality is constituted by several characteristics, such as sweetness, acidity, nutrient content, vitamin C (VC), and amino acid components. Approximately 50% of the dry matter is sugar, mainly including glucose, fructose and a small amount of sucrose, which determines the level of soluble solids. Approximately 12% of the dry matter comprises organic acids, including citric acid and malic acid, which determine the sourness of fruits [10,11]. Due to the limited root volume and low ion buffering capacity, the amount of irrigation water and its nutrient content must be carefully controlled [7,12].

In soilless cultivation, fertilizers are supplied as ions in the nutrient solution, providing nutrients for plants [13]. Giving too much fertilizer in pursuit of high yield results in weaker plant growth, lower water and fertilizer efficiency, worse quality and accumulation of nitrate in the tomato fruit [14–16]. Therefore, nutrient solution concentration management is the key technology of substrate cultivation. To enhance plant growth and nutrient uptake, many formulations of essential macro and micronutrients have been developed [17,18]. The composition and concentration of these formulations are based on plant species and cultivar, experience and the growing system [19,20]. When the content of a nutrient element is too high, it will cause ion imbalance and produce toxic effects. Ammonium ($NH_4^+$) is toxic to cells when it is present at high concentrations in the nutrient solution because it causes the so-called 'ammonium syndrome' [21]. Excessive concentrations of sodium ($Na^+$) in the nutrient solution depress the functioning of cell membranes and metabolic activity and even cause cell death [22]. The density of root hairs was increased by a relatively high concentration of nitrate or a relatively low concentration of phosphorus (P), while P deficiency led to a decrease in primary root growth and an increase in lateral root growth [23,24]. During the various growth and development stages, plants need and take up specific concentrations of elements [25,26]. The appropriate nitrogen concentration of the nutrient solution which promoted the growth and development of tomato plants was found to be 15 mmol·$L^{-1}$ Additionally, there was a high demand for nitrogen during fruit ripening due to the high level of free amino acids in tomato fruits [27,28]. Low and excessive concentrations of phosphorus lead to a decrease in photosynthetic capacity [29]. Elevated amounts of calcium are required in leaf growth periods, while potassium uptake is greatly increased at the full fruit load stage. A potassium concentration of 9 mmol·$L^{-1}$ during the flowering stage increased the soluble solids content of tomato fruits, but an excessive potassium concentration reduced yield. A high supply of potassium at the fruit ripening stage enhances the yield and taste of tomatoes but also increases the risk of blossom end rot [30,31]. In the vegetative stage, nitrogen (N) is important for chlorophyll formation and promotes the growth of plant height, leaf area and the number of flowers [32]. In the flowering stage, P (responsible for the number of flowers and buds formed) and potassium (K, which promotes flower initiation) are the most required elements. During fruiting, a large amount of K (which stimulates flowers to mature and form fruits) is needed [30–32]. Thus, a better understanding of plant periodical responses to water and fertilizer is important for nutrient solution management in practice.

Improving the quality of tomatoes while ensuring tomato yield has become a hot spot in research on vegetable production. In this context, management of the nutrient

solution electrical conductivity (EC) has been intensively evaluated as an effective strategy. Normally, the EC is kept in the range of 1.5 to 3.5 ms·cm$^{-1}$ to obtain the optimal yield [12]. It has been hypothesized that an increased EC level may be beneficial to plants due to an improved nutritional value. Some previous studies have shown that the fruit quality was significantly improved when the EC was increased from 1.5 to 2.4 ms·cm$^{-1}$ at the early stage of fruit development, while the yield per plant was not affected [31]. The lycopene concentration was enhanced when tomato plants were grown with a nutrient solution of 4.8 ms·cm$^{-1}$ compared to 2.4 ms·cm$^{-1}$ [33]. Irrigation with 4/3 times the concentration of a standard nutrient solution from the harvest of the first truss of tomatoes not only promoted the accumulation of dry matter but also improved the soluble sugar, VC and lycopene contents [34]. The short-term high-concentration nutrient solution before harvest can improve the content of glucose and fructose in the later stage of tomato fruit development [34,35]. Therefore, reasonable nutrient solution management in substrate cultivation is particularly important to improve tomato quality.

Solanaceous fruit vegetables have a long growth period with different demands for nutrients, and researchers mainly focus on the effect of a single nutrient solution concentration on their growth and development. Additionally, the results are inconsistent with different crops and varieties. Very little investigation has been carried out on cherry tomatoes supplied with different concentrations of nutrient solutions at different growth stages. Therefore, our research aimed to define the most effective nutrient solution strength (within the EC range of 1.5–4.5 mS/cm) on the growth, yield and quality performance of cherry tomatoes grown in northern China and to provide a scientific theoretical basis for tomato nutrient solution under coconut cultivation.

## 2. Material and Methods

### 2.1. Site and Material

A variety of *Solanum lycopersicum* cv. Busan 88 was chosen as the experimental material. This variety has the following advantages: strong plant growth and environmental adaptability, and the fruit is neat and hard with good quality. The cherry tomato plants were grown in a single-span solar greenhouse located in Changping District, Beijing, China (40°22′ N 116°23′ E) from September 2021 to February 2022.

Coconut bran trough culture is adopted for planting, with a length of 8.2 m and a width of 0.4 m. It is planted in a single row with vines hanging on the left and right, double pole pruning, plant spacing of 20 cm, large row spacing of 140 cm, and a planting density of 40 plants per trough.

### 2.2. Experimental Design

Tomato seedlings at the four-leaf stage were planted in planting troughs and watered with different concentrations of nutrients after the rejuvenation period. The nutrient solution concentrate formulation is as follows: calcium nitrate tetrahydrate 1090 mg·L$^{-1}$, potassium nitrate 430 mg·L$^{-1}$, potassium dihydrogen phosphate 205 mg·L$^{-1}$, ammonium sulfate 33 mg·L$^{-1}$, magnesium sulfate heptahydrate 430 mg·L$^{-1}$, potassium sulfate 305 mg·L$^{-1}$, ferric sodium ethylenediamine tetraacetate 6.4 mg·L$^{-1}$, trihydrate manganous sulfate monohydrate 1.7 mg·L$^{-1}$, monohydrate zinc sulfate 1.5 mg·L$^{-1}$, sodium perborate tetrahydrate 6.3 mg·L$^{-1}$, copper sulfate pentahydrate 0.2 mg·L$^{-1}$, and sodium molybdate dihydrate 0.2 mg·L$^{-1}$. The prepared nutrient solution has a potential of hydrogen (pH) of 6.1–6.2 and electrical conductivity (EC) of 3.0–3.2 ms·cm$^{-1}$. The whole growth and development cycle of tomato was divided into three key periods for irrigation with different concentrations of nutrient solution. The 1st stage was defined as the seedling and flowering period from planting to the bloom of the third panicle. The 2nd was defined as the fruiting period from the flowering of the third panicle to the whitening of the fruit of the first truss. The 3rd was defined as the harvesting period from the whitening of the first truss to the end of the harvest. Combined with the relevant test results of our research group in the previous year, the later stage led to a reduction in yield. Therefore, on this

basis, we proposed five treatments in this experiment to optimize the management strategy of the nutrient solution (Table 1).

**Table 1.** Different concentrations of nutrient solutions based on different growth periods.

| Treatments | Electrical Conductivity (ms·cm$^{-1}$) | | |
| --- | --- | --- | --- |
| | Seedling and Flowering Period | Fruiting Period | Harvesting Period |
| CK | 3.0 ± 0.2 | 3.0 ± 0.2 | 3.0 ± 0.2 |
| T1 | 1.5 ± 0.2 | 3.0 ± 0.2 | 3.0 ± 0.2 |
| T2 | 1.5 ± 0.2 | 3.0 ± 0.2 | 4.5 ± 0.2 |
| T3 | 1.5 ± 0.2 | 1.5 ± 0.2 | 3.0 ± 0.2 |
| T4 | 1.5 ± 0.2 | 4.5 ± 0.2 | 4.5 ± 0.2 |

Three replicates were set up in this experiment, and all treatments within each replicate were randomly arranged in blocks. A barrel of 600 L was used to bottle nutrient solution at one side of the plant trough. A water pump of 55 W was installed at the bottom of the inner side of each barrel to distribute the nutrient solution. The nutrient solution was irrigated with two drip irrigation belts in each trough. The duration and frequency of irrigation were based on Pardossi (2011) et al. [36]. Fixed irrigation was performed four times a day, and drip irrigation was applied for 1–5 min each time according to plant growth and weather. All treatments were unified with double pole pruning, topping after seven spikes. Bumblebee pollination technology was used to ensure a proper fruit set.

*2.3. Methods*

2.3.1. Measurement of Plant Growth Index

Plant height was measured with tape from the base to the apical growth point of the plant. Stem diameter was determined with a Vernier caliper to measure the first spike position from the top of the plant. The number of leaves per plant was also determined. In this experiment, cherry tomato plants grew for nearly 6 months from planting to pulling after fruits were harvested (September 2021–February 2022), which were divided into three key growth stages (16 September–26 October; 26 October–26 November; 26 November–16 February). The growth indicators of tomato plants were measured in three stages before topping the plants.

2.3.2. Determination of Leaf Chlorophyll Content and Photosynthetic Parameters

The relative chlorophyll content was detected in five fully expanded leaves from each tomato plant using a SPAD502-plus handheld chlorophyll meter (Konicamino, Tokyo, Japan) to measure the intensity of light transmitted at 650 nm [37]. Photosynthetic parameters were monitored with a portable photosynthesis system LI-6400XT (LI-COR, Lincoln, NE, USA) as previously described [14]. The fully expanded mid-canopy functional leaves at the same leaf position were selected to measure gas exchange parameters before the end of stage 2. The light intensity was set at 800 μmol·m$^{-2}$·s$^{-1}$, and the level of $CO_2$ was set as 400 μmol · mol$^{-1}$.

2.3.3. Analysis of the Yield and Quality of Tomato Fruit

Ripe tomato fruits were picked and weighed, and the total yield was calculated as the sum of the weights of all fruits for each plant. At the same time, we calculated the equivalent yield per square meter [38]. The soluble solid content of the fruits was measured with a hand-held refractometer PAL-1 (Atago, Tokyo Japan) and expressed in °Brix as described by Hiranrangsee (2016) et al. [39]. The titratable acid content was obtained according to Ecuadorian standards. The maturity index (MI) was calculated based on the soluble solids content and titratable acidity content [40].

### 2.3.4. Assessment of Partial Fertilizer Productivity

The partial factor productivity of applied N (PFPN), P (PFPP) and K(PFPK) was calculated based on the ratio of fruit yield to fertilizer (N, P and K). Detailed measurement information can be found in Jiang (2021) et al. [41].

### 2.4. Data Statistics

The analysis of variance (ANOVA) was performed using statistics software 20.0 (IBM SPSS Inc., Armonk, NY, USA). The significant differences between treatments were evaluated at $p \leq 0.05$ significance levels with Duncan's multiple range tests. Single effect analysis was performed when the interaction was significant. The graphs were drawn with OriginPro software 2018 (OriginLab Corporation, Northampton, MA, USA).

## 3. Results

### 3.1. The Effect of Nutrient Solution Concentration Dynamic Management on Cherry Tomato Growth

After one week of planting, tomato plants with consistent growth potential were selected to measure plant height, stem diameter and number of leaves. As shown in Figure 1A, the height of tomato plants increased gradually over time. At the end of the first stage (seedling and flowering period, 10/26), the height of tomato plants supplied with a nutrient solution of 1.5 ms·cm$^{-1}$ (T1, T2, T3, T4) was significantly lower than that of the CK (EC = 3.0 ms·cm$^{-1}$). By the end of the second stage (fruiting period, 11/26), there were significant differences between the treatments and the control. The plant heights of T1, T2 (EC = 3.0 ms·cm$^{-1}$) and T4 (EC = 4.5 ms·cm$^{-1}$) were significantly higher than that of CK, while that of T3 was significantly lower than that of CK. Before topping in the third stage (harvesting period), T2- and T4-treated plants were still significantly higher than CK and other treatments, and there were no significant differences between them. T3 was significantly shorter than CK.

To better evaluate the effects of different nutrient concentrations on plant strength, stem diameter at the growing point was measured (Figure 1B). The stem diameter of tomato plants supplied with a 1.5 ms·cm$^{-1}$ nutrient solution (T1, T2, T3, T4) during the seedling and flowering periods was significantly thicker than that of CK. During the fruiting period, when plants were irrigated with 1.5 ms·cm$^{-1}$ nutrient solution (T3), the stem diameter was significantly thinner than that of plants irrigated with 4.5 ms·cm$^{-1}$ nutrient solution (T4). T3 and T4 showed no significant difference from the other treatments (CK, T1 and T2, EC = 3.0 ms·cm$^{-1}$). After the plant was supplied with a high-concentration nutrient solution during the harvest period (T2 and T4, EC = 4.5 ms·cm$^{-1}$), the plant stem diameter was thicker than that of the low-concentration nutrient solution treatment (CK, T1 and T3, EC = 3.0 ms·cm$^{-1}$). In addition, T2 and T4 were significantly thicker than T3.

By counting the leaves, we found that the tomato plants treated with a low-concentration solution in the first stage (T1, T2, T3 and T4, EC = 1.5 ms·cm$^{-1}$) had more leaves than CK (EC = 3.0 ms·cm$^{-1}$). In the second stage, T1 and T2 (both EC = 3.0 ms·cm$^{-1}$) had the most leaves, significantly more than other treatments, and the number of leaves of plants treated with T3 (EC = 1.5 ms·cm$^{-1}$) was the lowest, which was significantly lower than that of plants treated with T1, T2, and T4 (EC = 4.5 ms·cm$^{-1}$) but had no significant difference from CK (Figure 1C). In the third stage, all plants had more than 20 leaves, and the number of leaves of plants treated with T3 (EC = 3.0 ms·cm$^{-1}$) was significantly lower than that of the other treatment groups except T2 (EC = 4.5 ms·cm$^{-1}$). At this time, the leaf number of T2 and T4 was the largest, but the difference was not significant compared with CK.

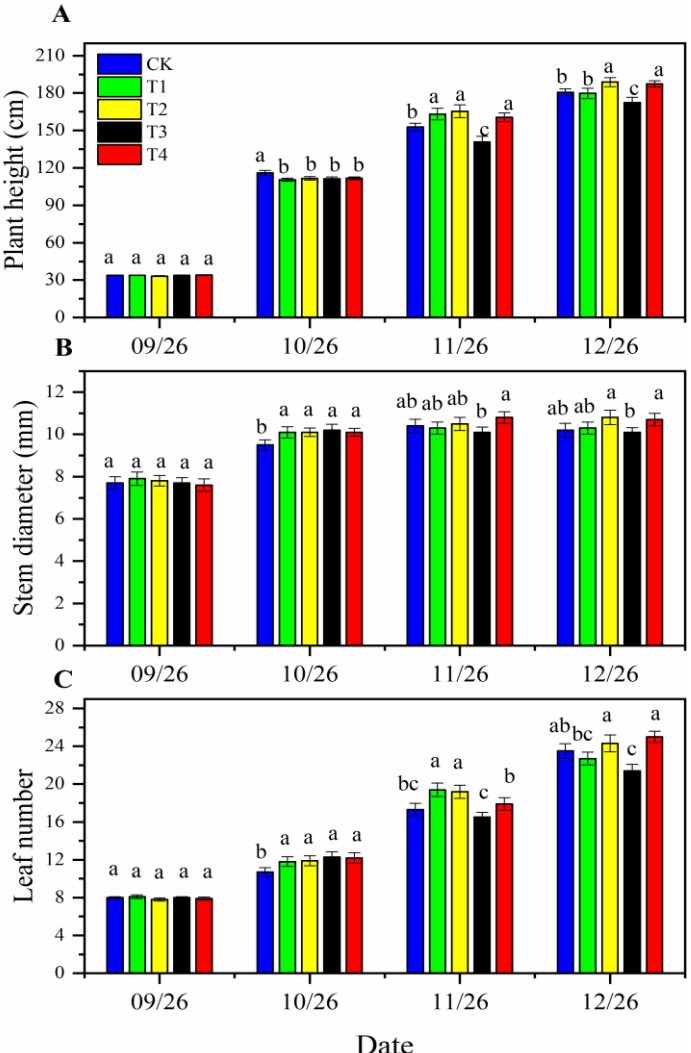

**Figure 1.** The growth of cherry tomato plants, supplied with different dynamic concentrations of nutrient solution. The plant height (**A**), stem diameter (**B**) and number of leaves (**C**) of tomato plants. The error bars represent the standard deviation (SD). Lowercase letters within the graph are used to compare individual mean values of the treatments, where having different letters indicates a significant differences ($p \leq 0.05$).

*3.2. The Effect of Nutrient Solution Concentration Dynamic Management on Chlorophyll Content and Photosynthesis in Cherry Tomato Leaves*

The chlorophyll relative content (SPAD) could reflect the photosynthesis level to some extent and could further affect plant growth. The results are presented in Table 2, Compared to CK (EC = 3.0, 3.0, 3.0 ms·cm$^{-1}$), when plants were given nutrient solutions, such as T2 (EC = 1.5, 3.0, 4.5 ms·cm$^{-1}$) or T4 (EC = 1.5, 4.5, 4.5 ms·cm$^{-1}$), the chlorophyll content increased. Although T3 (EC = 1.5, 1.5, 3.0 ms·cm$^{-1}$) showed no significant difference between CK and T1 (EC = 1.5, 3.0, 3.0 ms·cm$^{-1}$), it was significantly lower than T2 and T4. In addition, the net photosynthetic rate (Pn) and intercellular carbon dioxide concentration (*Ci*) of the photosynthesis parameters showed the same trend as the chlorophyll content. T4 and T2 had the strongest *Pn* and *Ci*, followed by T1 and CK, while T3 showed the lowest level of these two indicators. Similar to the transpiration rate (*Tr*) and stomatal conductance (*Gs*), there were some differences between all treatments for these two indicators, but they did not reach a significant level.

**Table 2.** Chlorophyll content and photosynthetic performance of tomato leaves supplied with different dynamic concentrations of nutrient solution. *Pn*, the net photosynthetic rates; *Tr*, transpiration rate; *Gs*, stomatal conductance; *Ci*, intercellular $CO_2$ concentration.

| Treatments | Chlorophyll Content (SPAD) | *Pn* ($\mu mol \cdot m^{-2} \cdot s^{-1}$) | *Tr* ($mmol \cdot m^{-2} \cdot s^{-1}$) | *Gs* ($mol \cdot m^{-2} \cdot s^{-1}$) | *Ci* ($\mu mol \cdot mmol^{-1}$) |
|---|---|---|---|---|---|
| CK | 37.6 ± 1.2 [ab] | 16.1 ± 1.3 [ab] | 6.3 ± 0.6 [a] | 0.42 ± 0.21 [a] | 264.8 ± 25.3 [ab] |
| T1 | 37.2 ± 1.1 [ab] | 16.5 ± 1.2 [ab] | 6.1 ± 0.2 [a] | 0.47 ± 0.16 [a] | 269.5 ± 19.5 [ab] |
| T2 | 38.1 ± 0.9 [a] | 17.7 ± 0.7 [a] | 6.8 ± 0.7 [a] | 0.48 ± 0.11 [a] | 274.8 ± 20.8 [ab] |
| T3 | 35.7 ± 1.1 [b] | 14.1 ± 1.3 [b] | 6.4 ± 0.3 [a] | 0.45 ± 0.24 [a] | 250.7 ± 18.5 [b] |
| T4 | 38.7 ± 1.2 [a] | 18.1 ± 1.3 [a] | 7.0 ± 0.7 [a] | 0.52 ± 0.08 [a] | 284.7 ± 15.8 [a] |

Note: Lower letters in the upper right corner of SD values represent significant differences ($p \leq 0.05$).

### 3.3. The Effect of Nutrient Solution Concentration Dynamic Management on Fruit Weight of Cherry Tomato

To determine the effects of dynamic nutrient solution concentration on tomato production, fruit yield per plant was investigated. The results showed that compared with CK, the yields of T1, T2 and T4 increased by 2.1%, 7.3% and 13.4%, respectively. T4 and T2 were significantly higher than CK, whereas no differences were revealed between T1 and T2 or between T1 and CK. However, the yield of T3 was significantly decreased by 6.9% (Figure 2A). Figure 2B shows that the order of average fruit weight was T4 > T2 > T1 > CK > T3, of which only T4 was significantly higher than CK and T3; there was no difference between the other treatments. For the average yield per square meter, T4 and T2 had higher production, which was significantly higher than that of CK and T3. Meanwhile, T3 was significantly lower than the other treatments except for T1. Tomato counting is the basic step for yield estimation. As shown in Figure 2D, there was little difference in the number of fruits per square meter among treatments, among which T3 was significantly lower than CK and T2 but not T1 and T4.

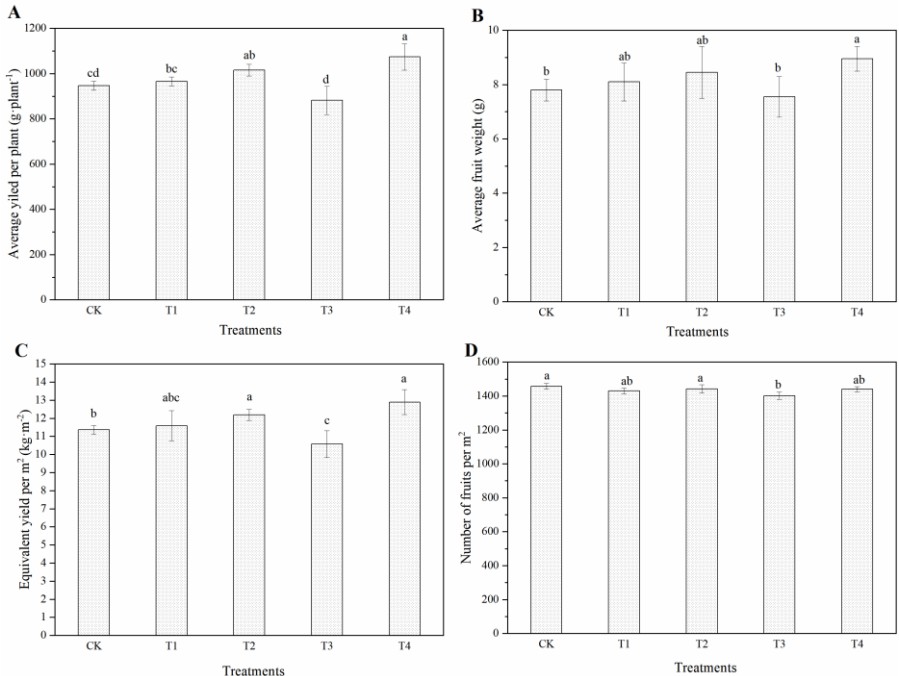

**Figure 2.** The yield of cherry tomato fruits supplied with different dynamic concentrations of nutrient solution. The average yield per plant (**A**), average fruit weight (**B**), equivalent yield per m² (**C**) and number of fruits per m² of cherry tomato fruits (**D**). The error bars represent the SD. Lowercase letters within the graph are used to compare individual mean values of the treatments, and different letters indicate significant differences ($p \leq 0.05$).

### 3.4. The Effect of Nutrient Solution Concentration Dynamic Management on Fruit Quality of Cherry Tomato

Comparisons of fruit total soluble solids (TSS) and total titratable acid (TTA) among the tomato fruit samples were examined to evaluate the quality of the fruits. The results in Table 3 show that different concentrations of nutrient solutions affected the quality of the tomato fruit in the trough experiments. The level of TSS was remarkably high in T2 and T4 fruit compared with CK fruit, which was increased by 9.1% and 9.8%, respectively. The TSS of T3 was the lowest and significantly lower than those of T2 and T4, while there was no significant difference between T3 and T1 and T2. All treatments had a certain effect on the level of TTA, among which the TTA of T2 and T4 decreased significantly, but the difference between the treatments was not significant. The maturity reflects fruit taste, as shown in Table 2, T2 and T4 significantly increased fruit maturity by 12.9% and 13.7%, respectively, compared with CK. T2 and T3 improved and deteriorated maturity, respectively, but the effect was not significant.

**Table 3.** The total soluble solids content, titratable acid content and maturity of cherry tomato fruit grown with different dynamic concentrations of nutrient solution. The average content of total soluble solids and titratable acid as well as fruit maturity from five harvests conducted on 24 Nov., 7 Dec., 20 Dec., 31 Dec. 2021 and 19 Jan. 2022. The mean and SD were calculated from three plants in each different growth trough. Lowercase letters in the upper right corner of the SD values represent significant differences ($p \leq 0.05$).

| Treatments | Total Soluble Solids (°Brix) | Total Titratable Acid (%) | Maturity |
|:---:|:---:|:---:|:---:|
| CK | 9.79 ± 0.29 [bc] | 1.74 ± 0.09 [a] | 5.99 ± 0.30 [b] |
| T1 | 9.85 ± 0.39 [abc] | 1.72 ± 0.09 [a] | 6.02 ± 0.30 [b] |
| T2 | 10.68 ± 0.53 [a] | 1.69 ± 0.08 [a] | 6.76 ± 0.34 [a] |
| T3 | 9.06 ± 0.45 [c] | 1.79 ± 0.09 [a] | 5.61 ± 0.28 [b] |
| T4 | 10.75 ± 0.54 [a] | 1.68 ± 0.08 [a] | 6.81 ± 0.35 [a] |

### 3.5. The Effect of Nutrient Solution Concentration Dynamic Management on Partial Factor Productivity of Cherry Tomato Production

Partial factor productivity (PFP) is an important indicator reflecting the comprehensive effect of fertilizer application. From the results of Table 4, by irrigating low-concentration nutrient solution in the early stage and high-concentration nutrient solution in the middle and later stages, the PFP of N, P and K were all improved. The T4 treatment had the highest PFPN, PFPP and PFPK, which were significantly higher than those of the other treatments except T2. T2 also had a strong PFPN, PFPP and PFPK, but the level of difference was not significant compared with CK and T1. The PFP of T3 was the lowest, while it was also not significantly different from CK. Dynamic management of the nutrient solution according to T4 (EC = 1.5, 4.5, 4.5 ms·cm$^{-1}$), T2 (EC = 1.5, 3.0, 4.5 ms·cm$^{-1}$) and T1 (EC = 1.5, 3.0, 3.0 ms·cm$^{-1}$) increased fertilizer partial productivity by 13%, 7% and 2%, respectively. The dynamic management of the nutrient solution according to T3 (EC = 1.5, 1.5, 3.0 ms·cm$^{-1}$) was ineffective, resulting in a 7% reduction in PFP.

**Table 4.** The partial factor productivity of different fertilizers on cherry tomato grown with different dynamic concentrations of nutrient solution (PFFN, partial factor productivity from applied nitrogen; PFFP, partial factor productivity from applied phosphorus; PFFK, partial factor productivity from applied potassium).

| Treatments | PFPN kg·kg$^{-1}$ | PFPP kg·kg$^{-1}$ | PFPK kg·kg$^{-1}$ |
|---|---|---|---|
| CK | 80.55 ± 4.83 [bc] | 675.47 ± 36.29 [bc] | 87.30 ± 4.74 [bc] |
| T1 | 82.17 ± 3.35 [b] | 689.02 ± 27.42 [b] | 89.05 ± 4.95 [bc] |
| T2 | 86.44 ± 3.76 [ab] | 724.55 ± 35.87 [ab] | 93.64 ± 4.41 [ab] |
| T3 | 74.99 ± 4.24 [c] | 628.82 ± 28.37 [c] | 81.27 ± 4.17 [c] |
| T4 | 91.34 ± 4.15 [a] | 765.99 ± 36.60 [a] | 99.00 ± 4.89 [a] |

Note: Lower letters in the upper right corner of SD values represent significant differences ($p \leq 0.05$).

## 4. Discussion

Nutrients are the key factors affecting plant growth and development. The vegetative and reproductive growth of tomato plants occurs simultaneously, which requires a great demand for fertilizer [42]. It was assumed that the regulation of nutrient solution concentration has an important effect on the growth of vegetables in soilless culture. A low concentration of the nutrient solution will slow the growth and development of the plant due to a lack of nutrients, while a high concentration of the nutrient solution will affect the absorption of nutrients and water because of osmotic stress around the root [43–45]. Therefore, the plants grew well under the appropriate concentration of the nutrient solution. This study also showed that in the early growth stage of tomato (mainly vegetative growth), properly reducing the concentration of the nutrient solution (EC = 1.5 ms·cm$^{-1}$) promoted an increase in stem diameter and leaf number (Figure 1B,C), which was beneficial to the vegetative growth of tomato. A nutrient solution with a higher concentration (EC = 3.0 ms·cm$^{-1}$) at this stage was not conducive to the plant growth and development and led to fewer leaves and slender stems, so it not only caused a waste of nutrients but also had an inhibitory effect. Similar effects on vegetative growth [46] as well as plant morphology have been reported [47]. The reduction in leaf area has been well documented in *F.x ananassa* corresponding to rapid plant adaptation to water deficit [48–50]; however, there was no difference in leaf area between the treatments in this study (data not shown). After entering the fruiting period, the amount of fertilizer absorbed by tomato plants increases sharply [51]. During this period, the demand for potash fertilizer and phosphate fertilizer is the greatest. Planters should pay attention to coordinating the relationship between leaves and fruits to promote the balance of sources and sinks. Additionally, the rational distribution of nutrients should be adjusted to strengthen the growth of leaves to achieve a high yield [52]. Under the conditions of this experiment, a nutrient solution with a reasonably high concentration (EC = 3.0 ms·cm$^{-1}$) at the second stage was more conducive to plant growth, as reflected by the highest plant height and most leaves (Figure 1A,C), which was conducive to the transfer of plant dry matter to reproductive growth [53]. While too low a concentration of the nutrient solution (EC = 1.5 ms·cm$^{-1}$) led to insufficient nutrient supply, or too high a concentration of the nutrient solution (EC = 4.5 ms·cm$^{-1}$) also led to excess nutrients, the latter had fewer adverse effects on plant growth at this stage. Once the plants were in the harvesting stage (mainly reproductive growth), since the number of fruit hanging on the plant was the largest, the nutrient demand was also large, and the concentration of the nutrient solution determined the fruit quality. The results showed that a higher concentration of the nutrient solution (EC = 4.5 ms·cm$^{-1}$) resulted in stronger growth as well as better TSS and maturity (Figure 1 and Table 3); however, a nutrient solution concentration that was too low reduced tomato quality, which is consistent with previous research [54,55]. Other studies showed that high-concentration nutrient solution caused nutrient stress to plants, and plants would perform osmotic adjustment in response to nutrient stress to prevent the loss of turgor pressure and provide a drive for tomato cell expansion. Changes in cell wall elasticity and initial turgor, although

turgor was maintained at normal levels, resulted in weaker growth than in nonstressed plants [42,56,57].

An appropriate nutrient solution concentration is beneficial for increasing the chlorophyll content [58]. This study also concluded that increasing the concentration of nutrient solution at different growth stages of tomato plants could increase the chlorophyll content (Table 2). The chlorophyll content of T2- and T4-treated tomato plants was the highest, while the chlorophyll content was significantly decreased when treated with the lower nutrient solution concentration (T3 treatment). Chlorophyll is the basic pigment for plant photosynthesis, and its changes directly affect the photosynthesis of plants [33,59]. Under low nutrient stress at any stage of the tomato growth period, the leaf chlorophyll content was decreased, resulting in a reduction in the photosynthetic rate and stomatal opening [60]. With the increase in nutrient solution concentration, the net photosynthetic rate, transpiration rate and stomatal conductance of tomato leaves increased considerably [61]. In the current study, the $Pn$ and $Ci$ of T3-treated (EC = 1.5, 1.5, 3.0 ms·cm$^{-1}$) plants were significantly lower than those of T4-treated plants (EC = 1.5, 4.5, 4.5 ms·cm$^{-1}$). Moreover, the other treatments had no significant effects on $Pn$, $Tr$, $Gs$ or $Ci$ (Table 2). Given that continuous growth of tomato plants in a nutrient solution with T3 negatively impacts the leaf chlorophyll and photosynthesis, this nutrient solution concentration appears to be a low nutrient stress condition that would deter growth and photosynthetic activity. Nutrients are important factors affecting photosynthesis, mainly through the control of photosynthesis-related enzymes, leaf pigments and stomatal or nonstomatal factors. The results of this study also showed that $Pn$ and $Tr$ increased with increasing nutrient solution concentration, which was similar to the conclusion of previous studies, which may be because increasing the nutrient solution concentration could increase the chlorophyll content of leaves and thus promote the photosynthetic carbon assimilation ability of leaves [62–64]. Suitable nutritional conditions are beneficial to improve the ecological environment of the substrate, enhance the photosynthesis of tomato, and benefit the growth of tomato.

Fertilizer plays a key role in increasing the yield of vegetables. The use of a reasonable concentration of a nutrient solution in soilless cultivation can not only increase crop yield but also improve the effective utilization rate of nutrients [3,5,65,66]. In addition, pollution of the environment can also be reduced, which is a reliable way to realize sustainable agriculture [67]. Concentrations of nutrient solutions that are too high or too low are disadvantageous to the growth of plants, not only affecting their vegetative growth but also leading to adverse consequences, such as yield reduction [27,34,52]. Tomato plants grown in T2 and T4 nutrient solutions showed the highest yield, with a larger weight of single fruits being produced (Figure 2A,B). However, the number of fruits was not different compared to fruits that were grown in CK, T1, T2 and T4 (Figure 2D). The yield per plant of tomato increased with the increase of the concentration of the nutrient solution in the last two growth stages. Additionally, there was a clear decrease in the yield compared to control nutrient solutions with low nutrient solution (T4) in the first two growth stages. In fact, the observed decline in cherry tomato production was similar to that reported for round tomatoes [12]. Fan (2003) [68] and Sun (2012) [69] showed that the rational application of nutrient concentrations promoted the coordination of root system growth and aboveground growth, which was beneficial to the accumulation of dry matter in the fruit, thus increasing crop yield. Fertilizer partial productivity (PFP) is an important index to reflect the comprehensive effect of fertilizer application. It answers the question of how productive a cropping system is in comparison to its nutrient input [70]. Through the dynamic management of the nutrient solution and applying corresponding concentrations of nutrition according to the growth state of plants, the PFP of the fertilizer could be improved (Table 4). These results suggested an appropriate increase (from EC = 3.0 ms·cm$^{-1}$ to EC = 4.5 ms·cm$^{-1}$ at the middle and late growth stages or to EC = 3.0 ms·cm$^{-1}$ at the late growth stage) in nutrient solution concentration will increase yield instead of reducing yield.

The cultivation system and nutrient solution management are the key factors affecting the quality of greenhouse vegetables. The sugar and acidity in tomato fruit greatly influence

the flavor quality. Generally, the taste quality of tomato fruit is judged by factors, such as TSS, TTA, and maturity [71,72]. Studies by Bai (2019) et al. [73] showed that an increase in the concentration of nutrient solution could increase the content of organic acids in tomato fruits. Cai (2018) et al. [74] found that the soluble solids content and the sugar-acid ratio of tomato fruits first increased and then decreased with increasing nutrient solution supply. In this study, the quality of the fruits was significantly improved with the higher TSS in T2- and T4-treated plants, which is consistent with hydroponically cultivated tomatoes which have a higher sugar content due to increasing nutrient solution concentration during fruiting development [33]. Indeed, the higher nutrient availability corresponding to the nutrient solution concentration resulted in higher soluble solids. Similar effects of increased nutrient salt solution on most fruit quality indicators were reported by Amalfitano (2017) et al. [75]. Conversely, in other studies, the market quality of berries was adversely affected by increased salt concentrations in the nutrient solution [76]. The TTA of all treatments showed no significant differences (Table 3). The increase in nutrient solution concentration increased the osmotic pressure of the nutrient solution to a certain extent, inhibited the root absorption of water and promoted the accumulation of sugar in the fruit [77]. These results demonstrate that cherry tomatoes respond to relatively high concentrations of nutrient solution in a similar way to round tomatoes, but the acidification seems to be less pronounced, which is a desirable trait. Some studies have shown that the suitable ratio of sugar to acid (maturity) of fruit tomato is 6–10 [78]. In this experiment, except for T3 and CK, the maturity of T1, T2 and T4 were in the range of suitable ratios between 6 and 7. Therefore, increasing the concentration of the nutrient solution in the third stage of tomato development is of great significance to improve fruit quality.

Taken together, the nutritional requirements of plants vary with developmental stage [79]. Several studies have shown that changes in nutrient solution concentration influence plant growth characteristics and fruit quality [33,80,81]. A growth reduction is expected for tomatoes with an increase in nutrient solution from an EC of 1.5 ms·cm$^{-1}$ to 3.0 ms·cm$^{-1}$ in the seedling and flowering growth stages. The strong ability of leaf photosynthetic carbon assimilation will be maintained with an EC range of 3.0 ms·cm$^{-1}$ to 4.5 ms·cm$^{-1}$ in the nutrient solution during the fruiting growth stage. Finally, at the harvesting growth stage, a higher EC will be good for tomato fruit quality formation.

## 5. Conclusions

Compared to supplying a nutrient solution with a constant concentration of 3.0 ms·cm$^{-1}$ throughout the growth cycle for tomatoes, the decrease in nutrient solution concentration from 3.0 ms·cm$^{-1}$ to 1.5 ms·cm$^{-1}$ at the seedling and flowering periods could effectively promote the growth of tomato plants and enhance leaf photosynthetic capacity. The increase in EC from 3.0 ms·cm$^{-1}$ to 4.5 ms·cm$^{-1}$ from the fruiting period or from the harvesting period would significantly improve cherry tomato fruit quality and increase yield. Furthermore, this nutrient solution concentration management strategy can also increase fertilizer productivity.

**Author Contributions:** T.L., Formal analysis, Investigation, Writing—original draft, Writing—review & editing. H.Y., Conceptualization, Funding acquisition, Methodology. W.J. Project administration, Supervision. T.Z. and T.W., Investigation, Data curation. C.S., Resources, Data curation, Methodology, Visualization. All authors have read and agreed to the published version of the manuscript.

**Funding:** This research was funded by National Natural Science Foundation of China (32002115). The National Key Research and Development Program of China (2019YFD1001900). The National Key Research and Development Program of China (2021YFD1600300). China Agricultural Research System (CARS-23-B07).

**Institutional Review Board Statement:** Not applicable.

**Informed Consent Statement:** Not applicable.

**Conflicts of Interest:** The authors declare no conflict of interest.

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
