# Peer review of "Influence of the Electrical Conductivity of the Nutrient Solution in Different Phenological Stages on the Growth and Yield of Cherry Tomato"

_horticulturae, doi:10.3390/horticulturae8050378_

Round 1

Reviewer 1 Report

English needs to be revised, not so much…..

The title of the paper should be more focused

The writing of the paper does not fully follow the authors' guide

Line 9 – soilless cultivation maximizes the production but not the quality

During the abstract the font of text is different

In the many case the term of culture must change with the crop or crops because is the correct term

Line 51- define VC

Line 61, the references is not correct cited

Line 65 – is not presented the toxicity of nutrients  

We recommend to improve the abstract for tomato crop nutrition https://www.mdpi.com/2077-0472/11/4/292

Line 105 experimental material can change with biological material

Material and methods

In general, for plant experiments, it is recommended that the repeatability of the experiment be for 2 years

Line 113 …resulting in a planting density of 12 plants/m2 - …I think it's a miscalculation because of the average distance between different rows.

Line 132    . it is not mentioned what works were applied during the vegetation period

Line 138   I do not think that the number of leaves is important per plant because it is genetically determined but the leaf area was more important because it was correlated with its production and quality.

Line 147 - it is not mentioned in what stage of development of the tomato plants (BBCH) the physiological determinations were carried out

Line 163 p≤0.05

Line 173  - from a scientific point of view, it is much more correct for development stages to be marked after BBCH

Line 187 delete and….Both

Table 2 we recommend to use the Legend under table

Figure 3 - which represents a, b?

Line 370 cultivation mode change with cultivation system

Conclusions

All together must be deleted…..:)

Conclusions are to generally, must be restored

Author Response

Dear Mr/Mrs Professor:

Thank you very much for your valuable comments on our manuscript.
We have carefully considerd your revison comments and made extensive modifications which are not limited to the following contents:
1.We have sent the revised MS to a professional language service organization for English Editing to make the MS more understandable.
2.The title has been changes to "Influence of Electrical Conductivity of the Nutrient Solution in Different phenological Stages on the Growth and Yield of Cherry Tomato", which maybe more focused this time.
3.Line 9: We change the sentence to “......allowing for the precise control of plant nutrients to maximize the vegetable production of uniform fruits”
4.In abstract, the font of text is different: Thanks for your kind reminder, I have unified the font of the whole paragraph.
5.Line 59: we difice Vc
6.Line 66-100: The inappropriate reference was deleted. And we give some other content according to the recommend article.
7.Line 127: We changed the materials to material.
8.In fact, this is our experiment in the second year. The experiment in the last year divided the plant growth stage into four stages. On this basis, we combined the first and second stages to form the current three stages.
9.Thank you for correcting the mistake of12 plants/m2,this spelling should be deleted.
10.I gave more cultivation management information from Line 139 to Line 157.
11.In fact, we also investigated the leaf area, but that of the same leaf position are not significantly different between different treatments. So here we report the data of leaf number only.
12.The fully expanded mid-canopy functional leaves at the same leaf position were selected to measure gas exchange parameters before stage 2 end. All growth measurements were carried out at the end of the each growth stage before topping.
13.We changed it to p≤0.05
14.We changed it to "tomato plants grew for nearly 6 months from planting to pulling after fruits have been harvested (09/2021 - 02/2022), which were divided Into three key growth stages (09/16 - 10/26; 10/26 - 11/26; 11/26 - 02/16)"
15.We added the note information under table 2,lower-letters with in the graph are used to compare individual mean values of the treatments, different letters indicate significant differences (P≤0.05).
16.We have reorganized the writing structure and content of the conclusion. This MS may still have a lot of contents to revise, and I hope you will give me more advises to meets the requirements for publication.

Best Wishes!

Reviewer 2 Report

This study estimated the “The Effects of Nutrient Solution Concentration Dynamic Management on Plant Growth, Fruit Quality and Yield of Cherry Tomato”. Indeed, soilless cultivation is of special interest. There are a lot of comments that should be taken into account by authors, which I believe are significant and important aspects that need to be thoroughly addressed in authors revision.

The main concern is:

Abstract:

(1) Lines 18-27: The presentation of the main results should be carefully and completely revised.  

Introduction:

(2) Lines 40-43: Please put references.

(3) Lines 88-94: these sentences should be rephrased.

(4) The references in some parts of the introduction section are not appropriate and cited blindly. I think the introduction is poorly written and it is too wordy.

(5) Authors should incorporate latest references dealing with the same topic. (6) At the end of this section, authors should illustrate what hypothesis this investigation aimed to test. Moreover, to verify this hypothesis, mention to the measured parameters.

(7) Research purpose is not clear - please make it more obvious.

Material and methods:

(8) Lines 115-116: these sentences should be rephrased.

(9) Line 125: The authors should explain these three key periods by days (give the old of the treated-plants in every period).

(10) The authors should explain why they used these five different concentrations of nutrient solutions (according to what?). For example: why they in T1 start with 3.0 ms.cm-1 nutrient solution, however in other treatments they start with 1.5 ms.cm-1 nutrient solution.

(11) Another important point for consideration is the fact that authors do not mention to the experiment design (The major concern deals with statistics).

Results:

(12) Lines 168-172: should given in material and methods section not in results section.

(13) Figures 1 & 3 could be arranged using a common X-axis.

(14) Where is Figures 2?

(15) Line 270: From the results of table 4 not table 3.

Discussion:

(16) Unfortunately, the Authors do not mention some recent literature on this topic and do not compare their results with those of previously published papers. Indeed some of the results would deserve this and, in the Discussion, commenting them and comparing them with published results would be more interesting than summarizing the results a second time.

(17) Authors should discuss how their results fill the gap of previous studies.

(18) At the end of this section, all evaluated parameters should be well integrated and discussed.

Conclusion:

(19) The conclusion section must be rewritten. Authors should include specific results of their research, which extend the current state of knowledge. 

References:

(20) References used by authors are not the newest one; I would like to ask them to use the newest one.

Linguistic quality:

(21) A further revision of the language is recommended.

Author Response

Dear Mr/Mrs Professor:

Thank you very much for your valuable comments on our manuscript.
We have carefully considerd your revison comments and made extensive modifications which are not limited to the following contents:
1. In abstract part, we have revised the contents.
2. Lin63, we give two refereces 10-11.
3. We rephrased related sentences as shown from L102-L111.
4. We have send the revised MS to AJE for english editing, so this revised MS maybe better.
5. We have rewritten this part and give some latest references, Maybe I don't have a comprehensive understanding. If there is better literature, please guide and provide some better to us.
6-7.From L118-L126, We re-summarize the purpose of this study.
8. These sentences have been rephrased.
9. The 1st stage was defined as seedling and flowering period from planting to the bloom of third panicle.
The 2nd was defined as fruiting period from the flowering of the third panicle to the whitening of the fruit of the first ear.
The 3rd was defined as harvesting period from the whitening of the first ear to the end of the harvest.
10.Combined with the relevant test results of our research group in the previous year, the concentration of 1.5 did not affect the growth of plants in the early stage,
but the concentration of 1.5 in the later stage led to a reduction in yield.
Therefore, on this basis, we proposed the treatment of this experiment to optimize the management strategy of nutrient solution.
11.Line 159-L169:Our experimental design is 5 treatments, which are randomly distributed in 3 replicates, and each treatment is supplied with a 600L nutrient solution bottle. 
12.Good suggestion, we move it to 2.3.1 part.
13.Your suggestion is very good. We also did a version of the figures at first, but found that the dynamic changes of the growth of the plant development process could not be better presented, so we changed it to the current form of Figure 1.
14.Just figure 2, this is a mistaken.
15.Thanks for your reminder.We corrected them.
16-18.In this section, we have supplemented some of the discussion with your valuable comments, and the last paragraph also summarizes some of the previous results.
19. We have rewritten the conclusion.
20. I have added some references, maybe not comprehensive, but I have try my best, If you have good and newest ones, please tell me and I will add them and cite them correctly.
21. I have send the revised MS to AJE for english editing, hope this will be better. This manuscript may still have a lot of contents to revise, and I hope you will give me more advises, to meets the requirements for publication.

Best Wishes!

Round 2

Reviewer 1 Report

Accept in present form

Reviewer 2 Report

 The comments have been mostly addressed except:

English still requires to be improved.